# Generalized Scale Factor Calibration Method for an Off-Axis Digital Image Correlation-Based Video Deflectometer

**DOI:** 10.3390/s222410010

**Published:** 2022-12-19

**Authors:** Long Tian, Tong Ding, Bing Pan

**Affiliations:** 1School of Science, China University of Geosciences, Beijing 100083, China; 2Institute of Solid Mechanics, School of Aeronautic Science and Engineering, Beihang University, Beijing 100191, China

**Keywords:** off-axis digital image correlation, scale factor, calibration, displacement measurement

## Abstract

When using off-axis digital image correlation (DIC) for non-contact, remote, and multipoint deflection monitoring of engineering structures, accurate calibration of the scale factor (SF), which converts image displacement to physical displacement for each measurement point, is critical to realize high-quality displacement measurement. In this work, based on the distortion-free pinhole imaging model, a generalized SF calibration model is proposed for an off-axis DIC-based video deflectometer. Then, the transversal relationship between the proposed SF calibration method and three commonly used SF calibration methods was discussed. The accuracy of these SF calibration methods was also compared using indoor rigid body translation experiments. It is proved that the proposed method can be degraded to one of the existing calibration methods in most cases, but will provide more accurate results under the following four conditions: (1) the camera’s pitch angle is more than 20°, (2) the focal length is more than 25 mm, (3) the pixel size of the camera sensor is more than 5 um, and (4) the image y-coordinate corresponding to the measurement point after deformation is far from the image center.

## 1. Introduction

Deflection (vertical displacement) of civil engineering structures under external loads is an important indicator that reflects the mechanical behavior and safety state of structures. In-situ real-time deflection monitoring can provide the necessary basic information for structural safety assessment and health monitoring. Therefore, deflection measurement has become one of the most essential detection contents for the health monitoring of various engineering structures (e.g., bridge, truss, railway, and roadbeds). Recently, bridge deflection measurement methods based on computer vision [1,2,3,4,5,6,7] have been emerging with the rapid development of digital imaging devices and image processing algorithms. Compared with traditional contact methods for bridge deflection measurement (e.g., dial gauges, accelerometers, linear variable differential transformers (LVDT) [8,9,10]), computer vision-based deflection monitoring methods offer outstanding advantages of non-contact, remote, multi-point, and high-accuracy measurement in a time-saving and labor-saving manner.

The principles behind computer vision-based deflection monitoring methods are straightforward and easy to implement. The first step is to capture video images of a test structure using a digital camera. Then, image displacements of measurement points are tracked in a series of images using certain digital image processing algorithms (e.g., intensity-based template matching algorithms or feature-based image registration algorithms). Finally, image displacements are converted to actual physical displacements using the pre-calibrated scale factor (SF), which can be determined based on the geometry relationship between the image and the test structure. Recently, the powerful digital image correlation (DIC) [11,12,13,14] technique, which is widely used in the experimental mechanics community for full-field displacement and strain measurements, has been successfully used for motion tracking of video images of engineering structures with artificial targets or natural textures. Based on the state-of-the-art DIC algorithm, an off-axis DIC-based video deflectometer was established by the present authors. Although DIC can precisely detect subtle image displacements, accurate determination of the SF at each measurement point which directly determines the final accuracy of measured deflections remains a challenge. 

Due to the limitation of testing conditions for field structure deflection measurement in outdoor scenarios, the optical axis of the camera used in the video deflectometer is no longer perpendicular to the test structure surface, leading to an off-axis imaging configuration. In this circumstance, it inevitably forms a pitch angle for the camera to image conveniently. Additionally, the field of view (FOV) of the camera is usually very large in the outdoor environment. Consequently, the distance between the optical center of camera and each measurement point is different, resulting in different SF for each measurement point converting image displacement to physical displacement. Thus, it becomes the key problem to accurately determine the conversion factor from image displacement to physical displacement for a high-precision deflection measurement based on the off-axis DIC method.

In previous studies, Feng et al. [15] presented a uniform calibration method when the optical axis is non-perpendicular to the test object’s surface. However, this method does not consider the difference in the position and distance of each measurement point in the case of multipoint measurement. Instead, all the measured points adopt the same SF, which seriously affects the measurement accuracy of each point in the case of multipoint measurement. Later, Pan et al. [16] proposed an approximate calibration method for off-axis DIC to obtain accurate measurement results in remote measurement with a fixed focal length of 50 mm without considering the effect of the deformed position of each measurement on SF. Based on this method, Tian et al. [17] presented a full-field method to calibrate all the points on the same line only by measuring the distance of no less than three measured points. Recently, Wang et al. [18] developed a novel calibration method that fully considers the variations in pitch angles in the full-field, which potentially improves the accuracy in measurement. However, each step of this method requires time-consuming inverse trigonometric function calculation, which is not conducive to the real-time displacement calculation.

This study aims to develop an accurate and fast calibration method for bridge deflection measurement based on off-axis DIC. The transversal relationship between the proposed SF calibration method and three commonly used SF calibration methods (i.e., Feng’s method, Pan’s method, and Wang’s method) are discussed and linked. In the following, the principle of deflection measurement using the off-axis DIC method is briefly described. Then, an off-axis imaging model is presented, based on which the mathematical equation of SF at each measuring point is deduced. Next, a generalized SF calibration method with a laser rangefinder is proposed and verified by vertical translation experiments with a high-precision vertical displacement controllable platform. Finally, the application range of the proposed SF calibration method is discussed, and a generalized calibration method is pointed out.

## 2. Off-Axis Digital Image Correlation-Based Video Deflectometer

The principles and implementation procedures of the off-axis DIC-based video deflectometer are schematically shown in Figure 1. First, a camera with appropriate resolution and an optical lens with suitable focal length should be selected for clear imaging according to the size and distance of the test structure. In practical outdoor applications, the optical axis of the camera cannot be aligned perpendicular to the vertical plane of the test structure, thus leading to a pitch angle *β* of the camera with off-axis imaging. In addition, the measurement points on the structure must possess evident features with random grayscale variations when the DIC algorithm is used to measure structure deflection/displacement. For large engineering structures, cavities, bolts, rivets, traffic signs, etc., can be used as natural speckles. If the natural texture of the structure surface is not obvious, artificial targets (e.g., LED lamps) should be installed. Afterward, the deflection measurement can be performed according to the following four steps: (1) adjust camera position and lens parameters to obtain a clear image of measurement points on the structure. Then, use one of the images (usually the image before loading) as a reference image, in which discrete pixel points are specified as points of interest (POI); (2) calibrate the SF for each POI; (3) real-time track the position of each measurement point in the image sequence captured after loading to obtain image displacement using subset-based DIC with sub-pixel accuracy; and (4) convert the image displacement of each measurement point to physical displacement according to the SF of each point obtained in step 2 and show the measurement result in real-time. In above steps, accurate determination of SF and high-precision tracking of image displacements are considered as the two key techniques in off-axis DIC-based video deflectometer for measuring structural deflection.

DIC is a well-developed and popular optical technique widely used in the experimental mechanics community for sub-pixel accuracy image displacement measurements. The basic principle of the subset-based DIC algorithm is straightforward. Namely, Reference square subsets centered at POIs are selected in the reference image to trace their positions in deformed images. Generally, a cross-correlation (or least square distance) criteria is used to evaluate the degree of similarity (or difference) between the reference subsets and deformed subsets. Subsequently, the matching procedure is completed by searching for the extremum of the correlation function. Once the maximum of the correlation coefficient is detected, the position and shape of target subsets in the deformed image can be determined. The differences in the position of the reference subset center and the target subset center yield the in-plane displacement vector at each POI. As algorithm details of DIC have been well documented in previous studies [19,20], here we will focus on the model and method for accurate SF calibration.

## 3. Scale Factor Calibration Based on a Generalized Off-Axis Imaging Model

### 3.1. Accurate SF Calibration Method

In using traditional 2D-DIC for in-plane displacement/strain measurements, the camera optical axis must be aligned to be perpendicular to the test object surface. However, when using off-axis DIC for the deflection monitoring of large structures, it is difficult or impossible to align the optical axis of the camera perpendicular to the measurement structure surface due to the limitation of instrument erection position. For this reason, a pitch angle exists between the optical axis of the camera and the horizontal plane, despite that the roll angle of the camera that represents the rotation of the camera around the optical axis can be adjusted readily to be zero. In this case, the SF of each point is no longer a constant because of the different distance from each measurement point to the optical center of camera. A suitable calibration model is therefore needed to determine the SF to convert the detected image displacement of each point to the actual physical displacement.

To accurately determine the SF of each point, the geometric model for off-axis DIC is established based on the pinhole imaging principle, as shown in Figure 2a. For consistency with the computer vision imaging model, the central perspective model is employed. An image coordinate system *O*_c_xy in the unit of pixels is established with the camera sensor plane center *O*_c_ as the origin point, *x*-axis to the right, and *y*-axis downwards. With the optical center *O* as the origin point, the camera coordinate system *O-XYZ* in the unit of mm is also established with the *X* and *Y* parallel to *x* and *y* axes of the image coordinate system, and the Z axis points to the test structure along the optical axis of the camera. For clarity, Figure 2b plots the imaging relation of point *P*_1_ in Figure 2a according to ideal pinhole imaging model. In Figure 2b, *O*_c_(*x*_c_, *y*_c_) denotes the image center, *f* is the focal length of the camera lens, *l*_ps_ is the physical size of the pixel, and *L* is the distance from the optical center of the camera to the measurement point, which can be measured using a laser rangefinder; *β* is the pitch angle that can be measured by a laser rangefinder or a theodolite.

In Figure 2b, we assume that measurement point *P*_1_ on the structure plane moves to *P*_2_ with a vertical displacement V. Correspondingly, point *p*_1_(*x*, *y*) in the camera sensor target moves to point *p*_2_(*x’*, *y’*), *v* is the image displacement in pixels. Since vertical displacement is the primary displacement component in structure deflection measurement, the horizontal image displacement is approximately zero. Based on the principle of trigonometric geometry, we have $\Delta Op_{1}p_{2}∼\Delta OP_{1}P_{2}\prime$, therefore
(1)$$\frac{\left|P_{1}P_{2}\prime \right|}{\left|p_{1}p_{2}\right|}=\frac{\left|OP_{1}\right|}{\left|Op_{1}\right|}$$

It can be seen from Figure 2b that the unknown parameter in Equation (1) can be expressed as:(2)$$\left|P_{1}P_{2}\prime \right|=\frac{\left|P_{1}P_{2}\prime \prime \right|}{\cos\beta \prime \prime }=\frac{\left|P_{1}P_{2}\right|\cos\beta_{2}}{\cos\beta \prime \prime }=\frac{V\cos\beta_{2}}{\cos\beta \prime \prime }$$
(3)$$\left|p_{1}p_{2}\right|=vl_{ps}$$
(4)$$\left|OP_{1}\right|=L$$
(5)$$\left|Op_{1}\right|=\sqrt{\left[{\left(x-x_{c}\right)}^{2}+{\left(y-y_{c}\right)}^{2}\right]l_{ps}^{2}+f^{2}}$$

By substituting Equations (2)–(5) into Equation (1) and after simplification, we obtain a relationship between *V* and *v*.
(6)$$K_{SF}=\frac{V}{v}=\frac{Ll_{ps}\cos\beta^{\prime \prime }}{\sqrt{\left[{\left(x-x_{c}\right)}^{2}+{\left(y-y_{c}\right)}^{2}\right]l_{ps}^{2}+f^{2}}\cos\beta_{2}}$$

It can be seen from Figure 2b that $\beta_{2}=\beta +\beta^{\prime \prime }$, where
(7)$$\tan\beta^{\prime \prime }=\frac{\left(y^{\prime }-y_{c}\right)l_{ps}}{\sqrt{{\left(x^{\prime }-x_{c}\right)}^{2}l_{ps}^{2}+f^{2}}}$$

Substitute Equation (7) and its trigonometric functions into Equation (6), and obtain the exact expression *K_SF_* as
(8)$$K_{SF}=\frac{l_{ps}}{\sqrt{\left[{\left(x-x_{c}\right)}^{2}+{\left(y-y_{c}\right)}^{2}\right]l_{ps}^{2}+f^{2}}}\times \frac{L}{\cos\beta }\times \frac{1}{1-\tan\beta \frac{\left(y^{\prime }-y_{c}\right)l_{ps}}{\sqrt{{\left(x^{\prime }-x_{c}\right)}^{2}l_{ps}^{2}+f^{2}}}}$$

To facilitate comparison, in the text below, we will use Equation (a) to replace Equation (8). As shown in Equation (a), the SF of measurement point P_1_ is related to the distance *L* from the optical center of the camera to the measurement point, the pitch angle of the camera *β*, the focal length *f*, the pixel size of the camera sensor *l*_ps_, the center of image coordination (*x*_c_, *y*_c_), and image coordinates of the measurement point before and after deformation.

In practical deflection/displacement measurement of engineering structures with off-axis DIC, *f* and *l*_ps_ are known parameters of the fixed focal length lens and the camera, *L* can be measured by a rangefinder, and *β* can be measured by the theodolite. Additionally, the image coordinate (*x*, *y*) of each measurement point is directly obtained after specifying measurement points in the reference image. Image coordinates (*x*, *y’*) of each measurement point in the deformed image can be computed using a subset-based DIC algorithm. Based on these parameters, the SF can be efficiently calculated using Equation (a) without affecting the real-time calculation performance of the video deflectometer.

### 3.2. Comparison with Existing Off-Axis SF Calibration Method

As mentioned earlier, various SF calibration models and methods have been established in previous studies. In this study, the transversal relationship between the proposed SF calibration method and three commonly used SF calibration methods is discussed.
(9)$$K=\frac{l_{ps}}{f}\times \frac{D}{\cos^{2}\beta }$$

In refs. [15], Feng and his coworkers established a simple imaging model. Based on this model, a uniform SF calibration method is developed as Equation (9), where *D* is the horizontal distance from the optical center of the camera to the vertical plane of the measurement points, which is not convenient to measure in practice.

Two distances are represented in Figure 3. Horizontal distance *D is* the length from the optical center *o* to the horizontal projection of the optical center *d*, the accurate position of *d* is difficult to determine in practice. Distance *L* is the length of the line from the optical center *o* to the measurement point *p*, which is very clear and easy to measure. To facilitate comparison, we will use Equation (b) to replace Equation (10). Equation (b) resembles Equation (9) but is simpler and more practical since the distance *L* is easier to measure than the horizontal distance *D*. As such, it can be called an improved Feng’s method.
(10)$$K_{SF1}\left(x,y\right)=\frac{Ll_{ps}}{f\cos\beta }$$

By considering the different locations of each measurement point, Pan et al. [16] proposed an easy-to-implement yet accurate SF calibration method for off-axis DIC-based video deflectometer as.
(11)$$K_{SF2}=\frac{l_{ps}}{\sqrt{\left[{\left(x-x_{c}\right)}^{2}+{\left(y-y_{c}\right)}^{2}\right]l_{ps}^{2}+f^{2}}}\times \frac{L}{\cos\beta }$$

To facilitate comparison, we will use Equation (c) to replace Equation (11). Compared with Equations (a) and (c), we can find that these two equations are also similar, and the difference is that a coefficient term is missing from Equation (c). The missing term considers the change in the pitch angle of every point, which is considered insignificant in Pan’s model. However, when *β* is not small, or *y*’ (i.e., y-coordinates) in the deformed image are far from the image center, the difference between Equations (a) and (c) would be significant.

More recently, Wang et al. [18] developed a model considering variations in the roll and pitch angles of every measurement point. The effect of the roll angle *α* is first considered as shown in Figure 4. The camera is rotated about the optical center $\left(x_{c},y_{c}\right)$ such that the roll angle is not zero. The relationship between (𝑥, 𝑦) and (𝑥*, 𝑦*) can be expressed as
(12)$$\left[\begin{array}{c} x*\\ y*\\ 1 \end{array}\right]=\left[\begin{array}{ccc} \cos\alpha & \sin\alpha & (1-\cos\alpha )x_{c}-y_{c}\sin\alpha \\ -\sin\alpha & \cos\alpha & (1-\cos\alpha )y_{c}+x_{c}\sin\alpha \\ 0 & 0 & 1 \end{array}\right]\times \left[\begin{array}{c} x\\ y\\ 1 \end{array}\right]$$

In Equation (12), a key parameter roll angle *α* must be measured in advance. In large FOV imaging, a small change in roll angle will not significantly affect the field of view. In other words, the roll angle is not necessary for imaging. Therefore, we can first level the camera so that the roll angle is zero, which can simplify the calculation. So, in the following steps, the authors gave two conclusions with and without roll angle.
$$\alpha_{0}=\arctan\left(\frac{\left(y_{c}-y\right)l_{ps}}{\sqrt{{\left(x_{c}-x\right)}^{2}l_{ps}^{2}+f^{2}}}\right)+\arctan\left(\frac{f\tan\beta }{\sqrt{{\left(x_{c}-x\right)}^{2}l_{ps}^{2}+f^{2}}}\right)$$
$$\alpha_{1}=\arctan\left(\frac{\left(y_{c}-y^{\prime }\right)l_{ps}}{\sqrt{{\left(x_{c}-x^{\prime }\right)}^{2}l_{ps}^{2}+f^{2}}}\right)+\arctan\left(\frac{f\tan\beta }{\sqrt{{\left(x_{c}-x^{\prime }\right)}^{2}l_{ps}^{2}+f^{2}}}\right)$$
(13)$$V=\frac{L\sin(\alpha_{0}-\alpha_{1})}{\cos\alpha_{1}}$$

In Equation (13), *V* is the vertical displacement of a point on the object, and *L* is the distance from optical center to measurement point. $\alpha_{0}$ and $\alpha_{1}$ are two available angles. If the roll angle *α* is not zero, the coordinates (*x*, *y*) in Equation (13) must be corrected by Equation (12). In Equation (13), the scaling factor SF reflecting the relationship between the actual displacement *V* and the image displacement *v* is not explicitly expressed. Without considering the roll angle, it can be known through mathematical derivation that it is equivalent to Equation (a). Equation (a) is more concise than Equation (13) in terms of mathematical form. Although the roll angle is completely avoidable in large FOV experiments, if there must be a roll angle, the coordinates should be corrected using Equation (12) before calculating *K*_SF_. To indicate the performance of the proposed SF calibration method, it is necessary to compare the proposed Equation (a) against the improved Feng’s Equation (b) and Pan’s Equation (c), as will be shown in the section below.

## 4. Experiments

To verify the accuracy and practicability of the accurate SF calibration method, in-plane vertical translation experiments were carried out in the laboratory using a video deflectometer and a high-precision vertical displacement platform. The displacement measurement results calculated by the proposed method are compared with the improved Feng’s method, Pan’s method, and the real displacement values. Furthermore, according to the experimental results, the applicable conditions of the three calibration methods are analyzed in detail.

### 4.1. Experiment Configuration

As shown in Figure 5, the video deflectometer consists of a high-speed monochrome camera (MER-131-210U3M, Daheng image vision co., LTD, China, resolution: 1280(H)×1024(V) with 8-bit quantization, maximum frame rate:210fps, pixel size: 4.8um), a fixed-focal length optical lens (the lens is replaceable as different measurement requirements), an optical theodolite, a Laptop computer (Thinkpad T440p, Lenovo, Intel(R) Core(TM) i5-4300M CPU, 2.60 GHz main frequency and 12 GB RAM) and a laser rangefinder (Leica DISTO D510, Leica Geosystems, Germany, distance measuring range: 200m, accuracy: ±1 mm).

The video camera is fixed onto an optical theodolite. By use of the optical theodolite, the pitch angle and horizontal (or yaw) angle of the video camera can be adjusted readily and then fixed tightly. The camera equipped with an optical imaging lens is connected to the laptop computer using a USB3.0 high-speed wire, through which the live video images of a test structure can be transmitted to the laptop computer and displayed in real-time. These live video images can be processed by the DIC algorithm described before to extract image displacement (in pixels) at specified measurement points, which can be subsequently converted into desired physical displacements (deflections) in millimeters based on the easy-to-implement and accurate calibration model described above. In the laboratory verification experiment, a high-precision translation stage was used to control the vertical translation of a measurement target with an accuracy of 0.01 mm.

### 4.2. Laboratory Translation Experiments

#### 4.2.1. Experiment Setup

A picture of the experiment setup for examining the accuracy of these SF calibration methods is presented in Figure 6. The test structure was a 250 mm × 250 mm plate, and the cross markers were pasted to 9 different positions of the test structure as measurement points. In this way, the laser rangefinder can locate the measurement points and obtain the distance of each point. At the same time, we can locate these measurement points in the image to obtain the corresponding image coordinates of each point. The test structure was fixed on the high-precision vertical displacement platform, which was placed vertically and fixed on a vibration isolation platform. The video deflectometer was placed on the ground to shoot the surface of the test structure with a pitch angle. The bubble of theodolite was adjusted to make the roll angle almost equal to zero.

During the experiments, two optical lenses with focal lengths of 8 mm and 50 mm were used. The optical lens and pitch angle of the camera were adjusted to obtain a clear image of the test structure and ensure the test structure was approximately in the center of the image. Then, the laser rangefinder was used to obtain the distance *L** from the camera sensor plane to each measurement point. Because the position of the optical center is inside the system which is invisible, the position of the camera plane is visible and easier to determine. The distance *L* from the optical center to the measurement point equals *L** minus focal length *f*. To reduce the measurement error, the distance measurement for each point was taken several times and the mean distance was calculated. The measured parameters are listed in Table 1.

The displacement platform was controlled automatically to make the test structure move down vertically 2 mm and stay for 15 s. After three times down repeatedly, the test structure was moved up by 6 mm to return to the starting position. The image displacements at each measurement point were computed by self-developed DIC software in real-time.

#### 4.2.2. Experimental Results

The image displacement-time curves of nine measurement points for lenses using two different focal lengths are shown in Figure 7. It is seen that the image displacements of these point on the test structure with only rigid body translation are not all the same. It is caused by the fact that the optical axis of the camera is not perpendicular to the surface of the test structure and the different distances from the optical center to each measurement point. In off-axis conditions, the SF of each point in the image is distinctly different.

Due to rigid body translation, the physical displacement of each point is the same. However, it can also be seen that the overall image displacements present a trend of gradual increase from top left to bottom right without SF correction. When the vertical displacement platform moved 6 mm, the difference between the maximum image displacement and the minimum image displacement reaches two pixels with 50 mm focal lengths. If a single uniform SF is adopted, there will be an obvious measurement error.

The physical displacements (deflection) converted from image displacement with improved Feng’s method, Pan’s method, and the proposed method are shown in Figure 8. As revealed by the figure, the deflection of each measurement point obtained with Pan’s method and proposed calibration method is highly consistent with the actual value, while the displacement of each measurement point obtained by improved Feng’s method in Equation (b) is inconsistent with a maximum difference of 0.43 mm.

Table 2 shows the root mean square error (RMSE) of each point for the proposed method in the case of 6 mm displacement for two different focal lengths. As shown in Table 2, the RMSE of each point is less than 0.06 mm. It proves the accuracy of the proposed off-axis calibration method.

#### 4.2.3. Discussion

Through the experimental results, the proposed calibration method shows good performance. However, it is also basically consistent with Pan’s method. Although the proposed model does not affect the real-time computation of the deflection measurement, the calculation is a little more complex than Pan’s method. In principle, the proposed method is more comprehensive, but from the experimental results, the method is not significantly better. Thus a detailed analysis of their applicable conditions should be discussed.

Equations (a)–(c) reveal an obvious linear relationship between the scale factor *K_SF_* and distance *L*. For quantitative analysis, the relationship between *K_SF_* and other parameters such as focal length *f*, pixel size *l_ps_*, pitch angle *β*, and the image coordinate of the measurement point (including the image coordinate before and after deformation), the common camera and lens parameters are selected to simulate the proposed calibration model shown in Equation (a) *K_SF_* and compare the model as shown in Equation (b) *K_SF1_* and Equation (c) *K_SF2_*.

We assume that the distance *L* is 100 m and the image resolution of the camera is 1280 × 1024 pixels. Generally, the pixel size of the camera is 3 µm to 10 µm. Five common focal lengths (8 mm, 16 mm, 25 mm, 50 mm, 100 mm) of the lens are selected. These parameters can meet the requirements of most of the field bridge deflection measurements. In the field bridge deflection measurement, the measured bridge can be adjusted to a position near the center of the image by adjusting the pitch angle of the camera. Therefore, the point (690, 562) near the center of the image is selected for comparative analysis. For simplicity of calculation, the displacement of the target point is assumed to be zero; that is, the position before and after deformation is the same. In Figure 9a, the pixel size of the camera is assumed to be 8 µm, and in Figure 9b, the focal length of the lens is 8 mm.

Figure 9 shows the influence of the camera’s pitch angle, the focal length of the camera lens, and pixel size on the SF. As can be seen, the SF increases with the increase in the camera’s pitch angle, the decrease in the focal length of the camera lens, and the increase in pixel size. The approximate *K_SF1_* calculated by Equation (b) and the approximate *K_SF2_* calculated by Equation (c) are almost the same. The difference between the proposed accurate calibration method and the approximate calibration method is almost negligible when the camera’s pitch angle is less than 20°, the focal length of the camera lens is greater than 25 mm, or the pixel size is smaller than 5 µm.

To analyze SF variation with the image coordinate (especially y-coordinate) of measurement points before and after deformation, full field SFs of an image with 1280×1024 resolution are fitted by selecting a point every 20 pixels. In this simulation, the distance *L* is 100 m, the focal length *f* is 50 mm, the pitch angle *β* is 20°, and the pixel size of the camera *l_ps_* is 4.8 um. The simulated result of full-field SFs before and after deformation (vertical translation 100 pixels) for the proposed *K_SF_* are shown in Figure 10, and the *K_SF2_* before deformation is shown. The *K_SF_*_2_ after deformation is not shown in Figure 10 because it equals to the SF before deformation. The *K_SF_*_1_ in Equation (b) is not calculated because it does not depend on the image coordinate and it always equals to the approximate SF in Equation (c) when (𝑥, 𝑦) is at the center of the image.

As shown in Figure 10, there is a big difference in the distribution of full-filed SFs between the proposed *K_SF_* and Pan’s *K_SF2_*, and *K_SF2_* is relatively small. For accurate calibration, SF is greatly affected by the image y-coordinate of the measured point, but less affected by the image x-coordinate. Additionally, it is affected by the position of the measured point after deformation. The SF is the smallest when the y-coordinate is at the center of the image after deformation. For approximate calibration, the influence of the x-coordinate and y-coordinate on *K_SF2_* is almost the same. The *K_SF2_* of the measurement point at the center of the image is always the largest, no matter before or after deformation. It can be seen from Figure 10d that the difference between *K_SF_* and *K_SF2_* is almost zero when the y-coordinate of the measured point moves to the image center after deformation. It indicates that the *K_SF_* almost equals the *K_SF2_* when 𝑦’ is in the middle position.

## 5. Conclusions

A generalized scale factor (SF) calibration method is proposed to determine accurate SF value at different measurement points for off-axis DIC-based video deflectometer. The relationship between the proposed SF calibration method and three commonly used SF calibration methods was analyzed and discussed. Indoor translation experiments validated the accuracy and practicality of the proposed SF calibration method and the existing Pan’s method. 

Although the results of these two calibration methods are similar, compared with Pan’s method, the proposed generalized SF calibration method considers the change in pitch angle when the measurement point moves to a new position in deformed images. This additional consideration of the proposed generalized SF calibration method provides more accurate results under following four conditions: (1) the camera’s pitch angle is more than 20°, (2) the focal length is more than 25 mm, (3) the pixel size of the camera sensor is more than 5 µm, and (4) the image y-coordinate corresponding to the measurement point after deformation is far from the image center. However, in other cases, the proposed method can be degraded or simplified to Pan’s SF calibration method with negligible difference but improved computational efficiency.

## Figures and Tables

**Figure 1 sensors-22-10010-f001:**
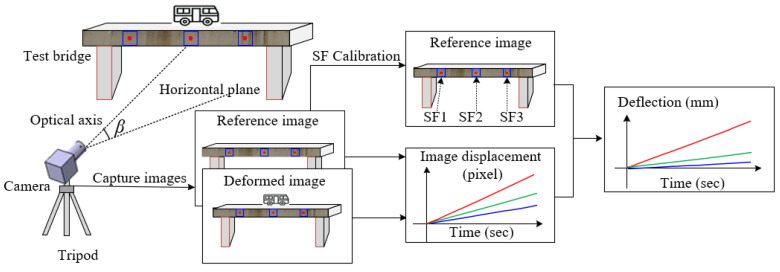
Schematic illustration of off-axis DIC-based video deflectometer for deflection monitoring.

**Figure 2 sensors-22-10010-f002:**
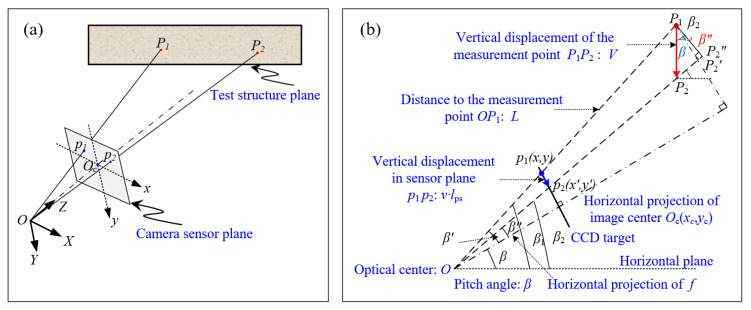
Imaging model of off-axis DIC: (**a**) geometric model with off-axis imaging of camera; (**b**) the off-axis imaging relation diagram of the measurement point.

**Figure 3 sensors-22-10010-f003:**
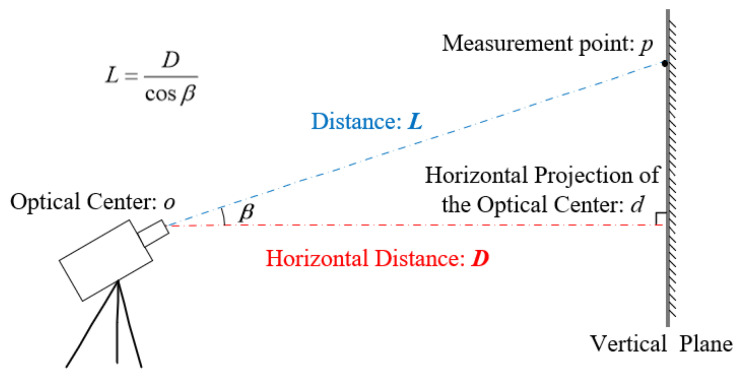
Schematic of the two distance representation methods.

**Figure 4 sensors-22-10010-f004:**
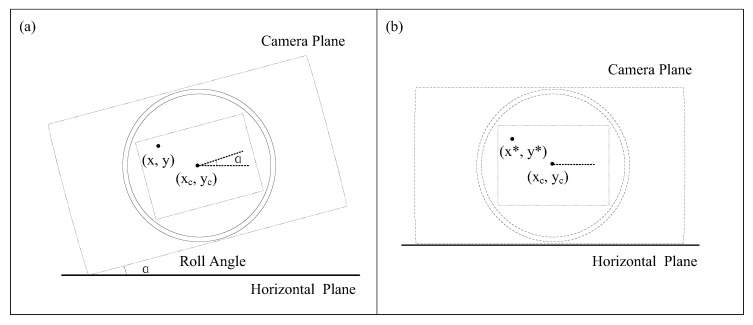
Schematic diagram of camera roll angle correction: (**a**) original model; (**b**) corrected model.

**Figure 5 sensors-22-10010-f005:**
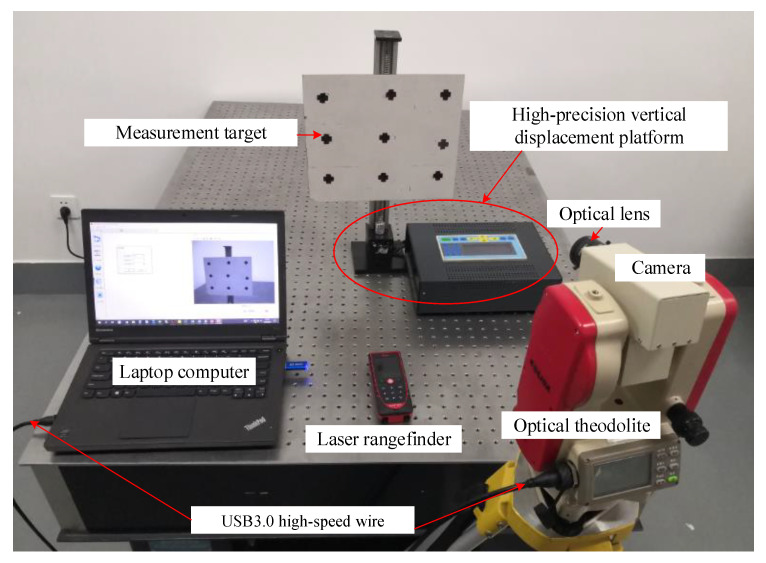
The video deflectometer and high-precision vertical displacement platform.

**Figure 6 sensors-22-10010-f006:**
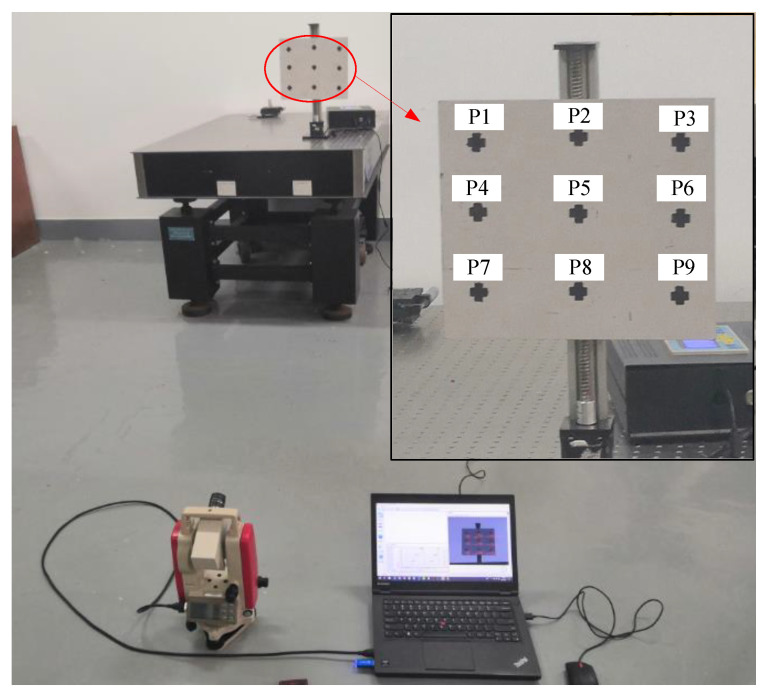
Experiment setup of laboratory verification tests.

**Figure 7 sensors-22-10010-f007:**
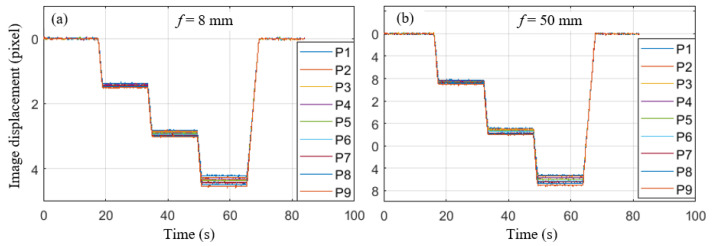
Image displacement of two camera lenses with different focal lengths: (**a**) *f* = 8 mm, (**b**) *f* = 50 mm.

**Figure 8 sensors-22-10010-f008:**
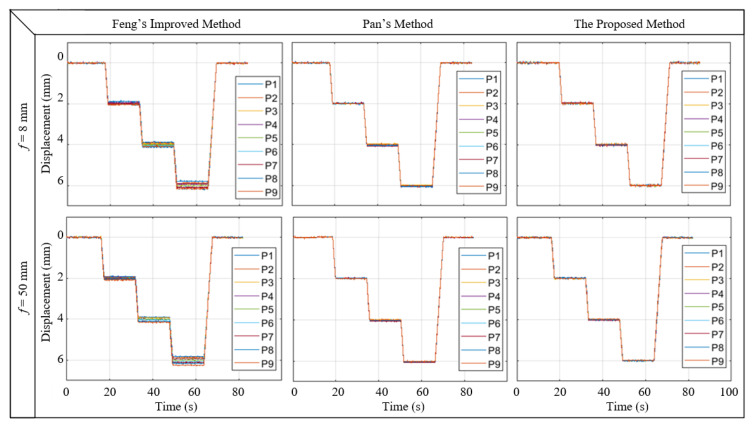
Displacement was calculated by three calibration methods and two different lenses.

**Figure 9 sensors-22-10010-f009:**
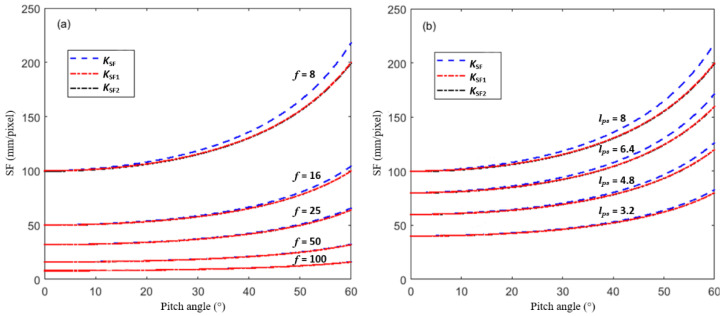
SF variation with the camera and lens parameters: (**a**) SF-pitch angle curve for different focal lengths of the lens, (**b**) SF-pitch angle curve for different pixel sizes.

**Figure 10 sensors-22-10010-f010:**
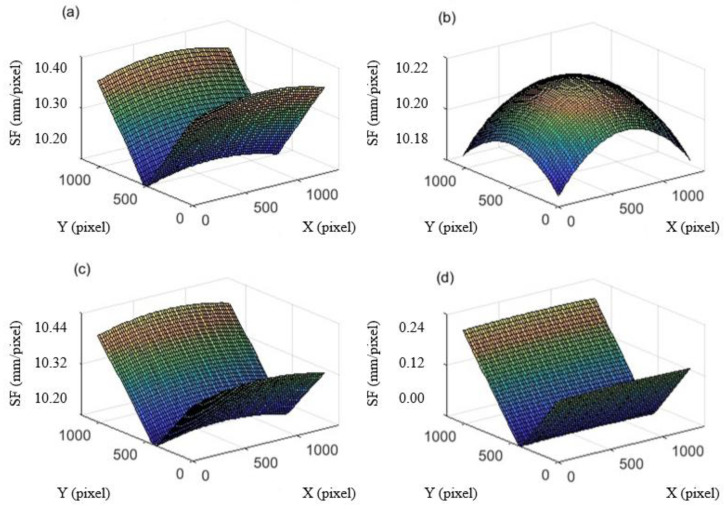
Simulated results of full-field SFs before and after deformation for two calibration methods: (**a**) the proposed calibration method before deformation, (**b**) Pan’s calibration method before deformation, (**c**) the proposed calibration method after vertical translation 100 pixels, (**d**) difference between the proposed and Pan’s calibration after vertical translation 100 pixels.

**Table 1 sensors-22-10010-t001:** Parameters for camera lens with different focal lengths.

Focal Length (mm)	Pitch Angle	Distance *L* (m)								
		P1	P2	P3	P4	P5	P6	P7	P8	P9
8	23.6°	2.284	2.280	2.276	2.257	2.251	2.246	2.226	2.221	2.216
50	22.1°	2.262	2.258	2.254	2.235	2.230	2.226	2.204	2.198	2.193

**Table 2 sensors-22-10010-t002:** RMSE of each point for the proposed method.

Focal Length (mm)	RMSE (mm)								
	P1	P2	P3	P4	P5	P6	P7	P8	P9
8	0.044	0.017	0.012	0.059	0.015	0.026	0.052	0.051	0.014
50	0.007	0.024	0.025	0.021	0.009	0.002	0.010	0.006	0.014

## Data Availability

The data presented in this study are available upon request from the corresponding author.

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
