# Peer review of "Generalized Scale Factor Calibration Method for an Off-Axis Digital Image Correlation-Based Video Deflectometer"

_sensors, 2022, doi:10.3390/s222410010_

Round 1

Reviewer 1 Report

This work proposes a generalized SF calibration model for an off-axis DIC-based video deflectometer based on the distortion-free pinhole imaging model. I found this work within the aim and scope of Sensors and would be happy to recommend its publication after my comments listed below are properly addressed by the authors:

(1)   It is suggested that the author modify the details of the figure in the paper, such as the blue line in Figure 1 and the image center coordinate in Figure 2.

(2)   It is suggested that the authors should explain the text "In equation (10), V is the vertical displacement of a point on the object and L is the distance. and are the two available angles, but they are expressed differently when there is or is not a roll angle." in a0 and a1 for further explanatory clarification, or modify the expression of this sentence.

(3)   Please check you mention the parameter.In the text " According to Eq. (9), the corrected ?*, ?*, ?*', ?*' corresponding to ?, ?, ?', ?' can be calculated if there is a roll angle. "The parameters ?*', ?*', ?', ?' that appear have not been explained by the authors, please add them.

Reviewer 2 Report

In this paper, a method for calibrating the scale factor (SF) of off-axis DIC is proposed, which considers the influence of camera pitch angle, the distance to target point and the position of the target point after deformation on the scale factor. But the paper needs to be revised to improve clarity and quality. The following issues need to be addressed:

1.     This paper introduces the influence of roll Angle and elevation Angle on SF, but does not mention the influence of yaw Angle on SF. However, yaw Angle sometimes occurs in the actual measurement process.

2.     Although the paper stated that the distance L was measured several times and averaged, because the optical center is inside the camera, the measurement of the distance from the target point to the optical center may have errors, and if the distance is close, this error cannot be ignored. If a better measurement method is used, a more specific description of how to use it should be given, the conditions under which the measurement error can be ignored can also be stated.

3.     It is suggested to indicate in what application scenarios the four situations mentioned in the conclusion will or will not occur, in order to demonstrate the advantages of the method proposed in this paper over the Pan’s method.

4.     Line 406, KSF and KSF2 should be KSF and KSF2, the manuscript should be reviewed throughout and corrected for similar mistakes.

5.     The meaning of ‘v’ should be illustrated in Figure 2 or earlier.

Reviewer 3 Report

I consider of interest the work developed, however I believe that the following changes should be taken into account for its improvement:

I recommend checking phrases like "Despite its high-accuracy registration accuracy, accurate…" (pp 2 line 50) and correcting mistakes, for example: “The measurement results of all points obtained by three calibration methods, the proposed method are consistent…”, removing the coma, since with comma "are" should be replaced by "is" (pp 13 line 400). V.

In the text, after Eq. (a3), it should be discussed what would happen when the change of the pitch angle of every point is not insignificant in Pans´s method.

The authors indicate in line 234 that, without  considering  the  roll  angle,  it  can  be  known  through mathematical derivation that Eq. (10) is equivalent to Eq. (a1). It is advisable to comment briefly what happens if the roll angle is not zero.

The authors indicate in line 288 that, to reduce the measurement error, distance measurement for each point was taken several times, and calculated the mean distance. Identify how many measurements have been made for each distance.

In pp 11, Tab. 2 shows the root mean square error (RMSE) of each point for the proposed method in the case of 6 mm displacement for two different focal lengths, indicating that the RMSE of each point is less than 0.06 mm in this method. Later, in the conclusion: “The measurement results of all points obtained by three calibration methods and the proposed method are consistent with actual displacement and the RMSE of every measurement point is less than 0.06 mm.”. This could lead you to believe that these 0.06 mm refer to the three methods. Clarify it in the text.

The conclusions should summarize more clearly the variables included in this new method compared to those included in the other three with which it has been compared, and not only when its application is more appropriate than the others.

Round 2

Reviewer 3 Report

I give a positive assessment in its new version. I express my congratulations for the research work carried out.